# Consensus for the Treatment of Tinea Pedis: A Systematic Review of Randomised Controlled Trials

**DOI:** 10.3390/jof8040351

**Published:** 2022-03-29

**Authors:** Harry Ward, Nicholas Parkes, Carolyn Smith, Stefan Kluzek, Richard Pearson

**Affiliations:** 1Versus Arthritis Centre for Sport, Exercise and Osteoarthritis Research, University of Nottingham, Nottingham NG7 2UH, UK; nicholas.parkes1@gmail.com; 2Bodleian Health Care Libraries, Oxford University, Oxford OX3 9DU, UK; carolyn.smith@bodleian.ox.ac.uk; 3Orthopaedics, Trauma and Sports Medicine, School of Medicine, Queen’s Medical, Centre University of Nottingham, Nottingham NG7 2UH, UK; stefan.kluzek@ndorms.ox.ac.uk (S.K.); richard.pearson@nottingham.ac.uk (R.P.)

**Keywords:** tinea pedis, athlete’s foot, treatment

## Abstract

Objective: To systematically review literature enabling the comparison of the efficacy of pharmaceutical treatments for tinea pedis in adults. Design: Systematic review of randomised controlled trials (RCTs) with mycological cure as the primary outcome. Secondary outcomes did include the clinical assessment of resolving infection or symptoms, duration of treatment, adverse events, adherence, and recurrence. Eligibility Criteria: Study participants suffering from only tinea pedis that were treated with a pharmaceutical treatment. The study must have been conducted using an RCT study design and recording age of the participant > 16 years of age. Results: A total of seven studies met the inclusion criteria, involving 1042 participants. The likelihood of resolution in study participants treated with terbinafine was RR 3.9 (95% CI: 2.0–7.8) times those with a placebo. Similarly, the allylamine butenafine was effective by RR 5.3 (95% CI: 1.4–19.6) compared to a placebo. Butenafine was similarly efficacious to terbinafine RR 1.3 (95% CI: 0.4–4.4). Terbinafine was marginally more efficacious than itraconazole, RR 1.3 (95% CI: 1.1–1.5). Summary/Conclusion: Topical terbinafine and butenafine treatments of tinea pedis were more efficacious than placebo. Tableted terbinafine and itraconazole administered orally were efficacious in the drug treatment of tinea pedis fungal infection. We are concerned about how few studies were available that reported the baseline characteristics for each treatment arm and that did not suffer greater than 20% loss to follow-up. We would like to see improved reporting of clinical trials in academic literature. Registration name: Treatment’s for athlete’s foot—systematic review with meta-analysis [CRD42020162078].

## 1. Background


**Summary**


What is already known?
Azoles and allylamines are superior to placebo in the treatment of tinea pedis;Allylamines may be superior to azoles, but the evidence is conflicting;The field has remained relatively unchanged since 1999.

What are the new findings?
There is huge inconsistency in RCTs for the treatment of tinea pedis, especially in terms of outcome measures;Many peer-reviewed RCTs for the treatment of tinea pedis have a high risk of bias, especially attrition bias;A common problem for these RCTs is the poor reporting of baseline characteristics for the treatment arms;Addressing these issues will streamline the application of future treatments for tinea pedis in clinical practice.

Tinea pedis, or athlete’s foot, is a superficial fungal infection of the skin of the feet caused by dermatophytes, most commonly *Trichophyton rubrum*, *Trichophyton mentagrophyte* and *Epidermophyton floccosum* [1,2]. Diagnosis is confirmed by the detection of segmented hyphae in skin scrapings using a potassium hydroxide (KOH) preparation and fungal culture from these skin flakes [3]. Increasingly rapid real-time diagnostic polymerase chain reaction (PCR) is being used, which not only allows the rapid identification of infection and the species, but also quantification, which can prove helpful in cases of recurrence of infection where there is clinical uncertainty over resolving pre-existing infection or recurrence [4].

Tinea pedis is common worldwide, with more than 70% of the population experiencing this infection during their lifetime [5]. Despite the advent of anti-fungal medication, incidence has increased in recent years [6]. Tinea pedis onychomycosis is more prevalent in older people [7]. Some groups are more at risk of dermatophyte infection due to occlusive footwear, sweating, and barefoot contact, such as miners, soldiers, and marathon runners [8,9,10]. Preventative adjunct treatment strategies include ensuring that the interdigital spaces remain dry, wearing well-ventilated shoes and socks composed of natural fibres, and covering feet when using communal facilities [11,12].

The first topical anti-fungal was developed in 1949. More recently, other compounds have been shown to have anti-fungal activity, such as tolnaftate, ciclopirox and undecylenic acid [11]. The list of compounds used to treat tinea pedis is expanding, with RCTs being used to assess the efficacy of herbal remedies [13]. While tinea pedis can be cured in most people, medical attention is often sought late. Even for those with treatment, tinea pedis can become severe and chronic. In those that have been successfully treated, reinfection is common and often does not self-resolve [1].

Prior systematic reviews assessing the treatment of tinea pedis include those within the Cochrane Database of Systematic Reviews. Crawford et al., 2007, concluded that placebo-controlled trials of topically applied allylamines and azoles for athlete’s foot consistently produce much higher percentages of cure than a placebo [12]. Then, in 2012, Bell-Syer et al. reported on oral treatments of tinea pedis; terbinafine is more effective than griseofulvin, and terbinafine and itraconazole are more effective than no treatment [14]. Not within the Cochrane systematic review structure, Rotta et al. reported in 2012 and 2013 their conclusions reached through systematic review regarding topical treatments [15,16]. Their manuscript indicated the superiority of topical antifungals over a placebo, but also that there is no consistent difference between classes. The efficacy of allylamines over azoles was implied in 1999 [17]; however, in this research, when allylamines and azoles were compared to a placebo, these two drug classes demonstrated similar efficacy. This superiority of allylamines over azoles was also further implied in the systematic review of the published literature [12,14,18], but has also been contested [15,16]. Given these conflicting results from previous RCTs and systematic reviews, there is a need for clarity in the field. We not only provide an update on this topic with the current evidence, but we also notice that many RCTs have a high risk of bias, particularly attrition bias, and often do not provide sufficient data on the baseline characteristics. We therefore employ careful inclusion criteria to exclude studies with high risk of bias that may have clouded previous systematic reviews.

Many now question the need for placebo-controlled trials when there is a known benefit of both classes of anti-fungal agents in treating tinea pedis [18]. Some consider that there is a lack of long-term data, e.g., 6-month follow-ups, to assess the efficacy of treatments on the recurrence rate of tinea pedis [14]. More recently, concerns regarding community *Trichophyton* resistance to terbinafine particularly in India have been raised [19,20]. The aim of this systematic review is to streamline current evidence for clinicians whilst highlighting areas for future research and troubleshooting problems in study design and interpretation so that future RCTs can be translated into clinical benefit.

## 2. Methods

This systematic review adheres to the PRISMA (Preferred Reporting Items for Systematic Reviews and Meta-Analyses) guidelines, including the use of the PRISMA checklist (see Appendix A TP PRISMA checklist). The search strategies for MEDLINE, Embase, Web of Science, Cochrane Database of Systematic Reviews, Cochrane Central Register of Controlled Trials, ClinicalTrials.gov, ICTRP, and LILACS were generated by CH, Bodleian Library (see Appendix A TP example search strategy Medline). Databases were searched on the 26th of May 2021. The initial dates for the literature search were database-dependent, for example, Ovid MEDLINE 1946, EMBASE 1974.

The selection of studies employed the conventional PICO approach. Participants were >16 years old with tinea pedis only treated with an accurately described pharmaceutical treatment. Intervention was described in detail, recording pharmaceutically active ingredients and formulations, including the dose and frequency of administration. The comparator, arm or arms that could have been a control treatment (placebo) were reported with the baseline characteristics of participants within each arm. The outcomes, mycological cure rates as determined by mycological culture and KOH microscopy, clinical cure, duration of treatment, adverse events, adherence to treatment, reoccurrence of infection, use and efficacy of adjuvants.

### 2.1. Inclusion Criteria

Participants were patients over 16 years old with only tinea pedis, well-described interventions, comprising of two different treatments, including the formulation, dose and frequency, a minimum of age and sex baseline characteristics reported in each arm, and outcomes reported as the mycological efficacy based on both the KOH and culture and reported separately to clinical data. Studies were required to follow the randomised controlled trial (RCT) experimental design within a hospital setting; however, registered trials by pharmaceutical companies also met the inclusion criteria. RCTs conducting intervention must be described in detail, recording the pharmaceutically active ingredient and formulation; for example, tablet or topical antifungal medications in the form of sprays, gels, creams, powders or ointments; including the dose and frequency.

### 2.2. Exclusion Criteria

A heterogeneous participants group i.e., not only tinea pedis, was an exclusion criterion, as was studies with participants < 16 years. The control group was required to be either a different treatment or placebo arm. We stipulated that the study must be excluded if the control arm or placebo group did not report the baseline characteristics; they must have included a minimum of the age and sex for each arm, and if there was more than 20% loss of participants in follow up.

### 2.3. Data Analysis

A total of 1850 study titles with abstracts were screened by two independent reviewers (NP and HW) for eligibility against the inclusion/exclusion criteria. This was followed by screening the full text of 140 full-text manuscripts. Disagreements between reviewers were resolved by consensus or by the decision of a third senior reviewer. Of the 1850 titles, 7 were deemed to meet the inclusion criteria and not fulfil the exclusion criteria (Figure 1).

Data were extracted using a data extraction proforma for each included study. This was used for the review and where the data were not suited to such an analysis; a text summary of the extracted data is described in the results section of this manuscript.

### 2.4. Risk of Bias Assessment

The Cochrane Risk risk-of-bias tool embedded within the ReviewManager (RevMan) software, V.5.4 (The Cochrane Collaboration, London, UK, 2020), was used to assess potential study bias. It was reported using a traffic light icon plot, where green represents low risk and red high risk of bias. Risk of bias analysis was performed over several domains: selection bias, performance bias, detection bias, attrition bias, and reporting bias. For selection bias, we assessed the methods of randomisation and allocation concealment. We assessed the blinding of participants and personnel for performance bias and blinding of outcome measures for detection bias. For attrition bias, we calculated the loss of participants from the baseline to the final outcome and we deemed a loss greater than 20% as high attrition bias. We then assessed reporting bias by comparing their proposed study design and their published data to determine if all of the outcomes were reported as intended from the outset of the study and also to determine if these outcomes were reported equally for both arms. We also assessed for any other bias, such as cross-over between arms.

## 3. Results

All of the included clinical studies were randomised controlled trials. Seven studies met the inclusion criteria while not meeting the exclusion criteria. Five involved topical application, whereas tablet formulations were used in two studies. These studies involved a total of 1042 participants, 680 not lost to follow-up, and had mycological data, which were both microscopy and cultures. The included studies were Smith et al., 1990 [21], Evans et al., 1991 [22], De Keyser et al., 1994 [23], Hay et al., 1995 [24], Syed et al., 2000 [25], Korting et al., 2001 [26], and Li et al., 2014 [27] (see Table 1). These studies were all clinical trials that were published in English. In general, the studies tended to be small in size, with between 10 and 35 participants within each study arm. De Keyser et al., 1994, Hay et al., 1995, and Li et al., 2014, were larger studies, with 88 to 184 participants in each study arm. Although these participant numbers were declared in the recruitment and demographic description of the study participants, the marked percentages of those constituting the study arms were excluded either due to factors such as negative mycology at the time of commencing treatment or attrition. This was most evident in the larger studies, where those not included in the analysis ranged between 23 and 133 within a study treatment arm.

### 3.1. Participant Characteristics

The reported mean or median age of the study participants ranged from 29 to 48 years for the placebo arms, 35 to 47 years for the terbinafine arms, and the itraconazole and butenafine arms having mean ages of 46 and 36 years, respectively. The proportion of males in the studies ranged from 54 to 80%, with the exception of the Syed et al., 2000, study being all male [25]. The infection status of all participants was confirmed by both culture and microscopy prior to the study, and any participants later found to be culture negative at baseline were removed from the analysis. All participants had clinical signs and symptoms of tinea pedis, although the studies varied in the signs and symptoms assessed. Several studies reported disease status on a four-point ordinal scale: absent, mild, moderate, or severe. The two disease criteria of erythema and pruritis were unanimously reported. Other commonly reported disease phenotypic characteristics included vesiculation (in eight out of nine audited studies), pustules (in seven out of nine), incrustation (in six out of nine) and scaling (in five out of nine). Erosions and macerations were only reported in one study.

### 3.2. Treatment and Outcome

The included studies tended to cover topical application formulations. All topical applications contained a 1% active ingredient formulation. However, De Keyser, 1994, and Hay, 1995, both trialled terbinafine (250 mg) against itraconazole (100 mg) as tablets, each being dosed once daily for 2 weeks. The duration of treatment varied across the trials, ranging from a single application of an alcohol-containing film-forming solution in the Li et al., 2014, study to a 4-week treatment period in the Smith et al., 1990, randomised controlled trial [21,27]. The duration of the study was variable, ranging from 4 to 16 weeks (Table 1). There was some variation in the timepoint chosen to assess the treatment efficacy. In the included studies, this ranged from the point when the treatment period ended, as in the Syed et al., 2000, study, up to 7 weeks after the completion of treatment, as reported by Korting et al., 2001 [25,26]. Mycological cure data derived from KOH microscopy and culture were used in our efficacy analyses, even if the paper chose to report in addition to the overall efficacy using a combination of clinical cure and mycological cure. Therefore, for a study to be included in this systematic review, it was a requirement that these data were reported separately in the published manuscript [21]. This was conducted in an effort to reduce inter-trial variability. Clinical cure data across the included studies did not refer to validated clinical assessment tools.

The specific clinical criterion used to define the resolution of the infection and hence the clinical efficacy varied between the RCT protocols for the different studies. The most common method was the sum of signs and symptoms on a 0–3 scale composed of absent, mild, moderate, and severe. However, not all studies used the same number of signs and symptoms. Some studies used percentages to gauge the amount of clinical improvement from poor, fair, good, excellent, to complete response, which corresponded to 0–25%, 25–50%, 50–79%, 80–99% and 100%, respectively. Some studies defined a ‘clinical cure’ as a total score of less than two on the 0–3 scale for each sign and symptom, whereas others defined it as less than 50% of signs and symptoms remaining, and some studies did not define it at all.

Each study was reviewed for the number of adverse events in each arm, adherence to treatment, recurrence of signs and symptoms and the use of any adjunct treatments/preventative strategies, such as advice on dry shoes and socks or specific insoles. No study precisely described how adherence to treatment was assessed, and no study addressed the benefit of adjunct treatments. The reviewed RCTs did not have experimental protocols designed to assess the recurrence rate.

### 3.3. Efficacy of Drug Treatment of Tinea Pedis

The likelihood of resolving tinea pedis infection in the study participants treated with terbinafine was 3.9 (95% CI: 2.0–7.8) times those treated with a placebo (Figure 2). Similarly, the allylamine butenafine was also more effective at treating tinea pedis by 5.3 (95% CI: 1.4–19.6) compared to a placebo. For comparison between butenafine and terbinafine, butenafine had similar efficacy to terbinafine 1.3 (95% CI: 0.4–4.4).

There were two two-arm studies that compared the oral administration of terbinafine versus itraconazole [23,24]. The tableted terbinafine and itraconazole analysis identified that terbinafine was 1.3 (95% CI: 1.1–1.5) more efficacious than the azole itraconazole.

Our analysis showed that five of the seven studies used in the analysis had a high risk of bias in one component of their study (Figure 3). This was predominantly regarding attrition bias, where incomplete outcome data were reported. One study illustrated a selective reporting bias. Although five of the seven studies reported one high risk of bias, there were a considerable number of occasions where there was an unclear risk of bias. This is indicative of a potential bias, as the bias topic received no or poor coverage in the published manuscript; therefore, the bias cannot be assessed.

## 4. Discussion

A total of seven studies met inclusion the criteria, involving 1042 study participants over the age of 16 years being treated for tinea pedis. A total of 680 participants had full mycological data reported. There were five studies assessing the efficacy of topical formulations, and two assessing tableted oral formulations. For topical formulations, the likelihood of resolving tinea pedis in study participants treated with terbinafine was RR 3.9 (95% CI: 2.0–7.8) times that of those treated with a placebo. Similarly, the allylamine butenafine was also effective at treating tinea pedis RR 5.3 (95% CI: 1.4–19.6) compared to a placebo. Butenafine was similarly efficacious to terbinafine by RR 1.3 (95% CI: 0.4–4.4). For tableted oral formulations: terbinafine showed increased efficacy compared with the azole itraconazole, RR 1.3 (95% CI: 1.1–1.5). The caveat regarding the analysis is the limited amount of data contributing to the analysis. This was principally due to our inclusion criteria stipulating that the participant demographics must be reported for each arm of the study. We chose to not combine topical applications with tableted formulations, as topicals are first-line treatments, whereas tableted formulations are second-line, or used if there is severe and extensive disease [28]. To summarise the certainty of our outcome in the review, we used the GRADE (Grading of Recommendations, Assessment, Development and Evaluations) scoring system [29]. This is a subjective evaluation of the certainty of an outcome from very low to high based on several factors, including risk of bias, imprecision, inconsistency, indirectness and publication bias. Given the limited number of studies, we were able to analyse and the high or unclear risk of bias in many of these studies, and we would rate the certainty of our outcomes as moderate.

### Common Pharmacological Treatment Groups for Tinea Pedis

Prior systematic reviews assessing the treatment of tinea pedis include those within the Cochrane Database of Systematic Reviews. Crawford et al., 2007, concluded that placebo-controlled trials of topically applied allylamines and azoles for athlete’s foot reduced the risk of tinea pedis treatment failure compared to a placebo [12]. Reporting risk in relation to treatment failure is counterintuitive from a clinical point of view; we took the more conventional approach of treatment success. Then, in 2012, Bell-Syer et al. reported on oral treatments of tinea pedis; terbinafine was more effective than griseofulvin, and terbinafine and itraconazole were more effective than no treatment [14]. Their analysis did not identify a difference between terbinafine and itraconazole, where we report a difference with RR 1.3 (95% CI: 1.1–1.5). In non-Cochrane systematic reviews, Rotta et al. reported in 2012 and 2013 the superiority of topical antifungals over a placebo, but also that there was no difference between classes [15,16]. Their data reported odds ratios and, due to the high prevalence in the numerator when making these calculations, the odds are not numerically comparable with the relative risk we report. The benefit of allylamines over azoles has been implied in systematic reviews [12,14,18], but also contested [15,16].

There are *a priori* relevant assumptions to consider when performing a systematic review; sensitivity, similarity, transitivity and consistency, respectively. Sensitivity analysis was not performed due to the limited number of included studies. Similarity addresses the methodology using the conventional PICO approach of population, intervention, comparison, and outcome. In this systematic review, as is usual, this was conducted initially when screening for eligibility through the inclusion criteria of the systematic reviews. Specifically, similarity was based upon four main aspects: the clinical characteristics of the study, treatment interventions, comparison treatments, and outcome measures. The clinical study design could vary to a degree, often regarding when the outcome assessment was scheduled. There was a range of signs and symptoms assessed, which were not consistent across the studies, making the extent of disease at baseline difficult to compare. Comparison was further confounded by the variety of methods used in different studies to rationalise these signs and symptoms into numerical correlates of ‘clinical cure’ after treatment. In this meta-analysis, the minimum requirement for inclusion was studies that reported baseline characteristics to ensure similarity between arms. We did not exclude studies where the attrition was due to negative cultures at baseline, although this may introduce a degree of selection bias. One study we excluded on the basis of selection bias as the population was previous non-responders [30]. Regarding the treatment interventions, the topical formulations were consistent in that they all contained 1% of active ingredient. The oral formulation doses differed between terbinafine and itraconazole, but were consistent for each drug and in line with prescription guidance. Topical formulations contained a range of excipients; the topical terbinafine formulation could contain alcohol to enable the terbinafine to enter the skin. Enabling the penetration of the active ingredient into the skin is likely to promote retention and hence affect the dynamics of exposure to the fungal infection. Factors such as this make it difficult to determine if the duration of exposure required to eradicate fungi differs between drug classes (allylamines and azoles). The adherence to these treatments was also never directly assessed in any of the studies. We ensured that, if the clinical assessment of tinea pedis was reported, it was in addition to mycology in the form of KOH microscopy and culture; we focused upon the eradication of the mycology associated with the disease. This was in part due to the lack of a validated clinical assessment protocol for tinea pedis and, hence, it could not be applied in the studies. On occasion it is likely for fungi to be present on the epidermis of feet where disease is not evident; similarly, symptoms associated with the disease may continue after the resolution of the infection—for example, nail changes in onychomycosis persist after eradication of tinea pedis [31]. In this review the comparison treatments did not present significant variation. Inconsistency and testing between direct and indirect treatments were not evident either globally (*p* = 0.42) or locally *p* > |z| 0.42.

#### Future Perspectives

In order to clarify the overwhelming literature in the field, the standardisation of RCT protocols for assessing tinea pedis treatment is required going forward, together with a standardised approach to reporting [32,33]. There needs to be a consensus on the clinical assessment of tinea pedis with a set list of signs and symptoms. This could be on the 0–3 scale (absent, mild, moderate, and severe), but needs to be validated so that it is reproducible across trials. All participants should have both a clinical and mycological diagnosis of tinea pedis at baseline, and the mycological diagnosis of tinea pedis should be defined as a positive culture and microscopy. Outcomes should be standardised so that they can be easily compared across studies. Mycological data for both culture and microscopy should be reported together at all time points so that the number of patients with mycological cure can be assessed. Ideally, PCR could be used to quantify fungal colonisation over time during treatment regimes. Studies should also accurately report the baseline demographics of the treatment arms and assess adherence to treatment, for example, by measuring the volume of topical ointment used at each visit and control this for the size (extent) of fungal infection.

Long-term data, e.g., 6-month follow-ups, to assess the efficacy of treatments on the recurrence rate of tinea pedis is a complex scenario [14]. This is due to the local foot environment. The role of the foot environment in the re-inoculation of the foot with fungi associated with tinea pedis is in question. Tinea pedis is associated with high rates of relapse. An effective anti-fungal should eradicate the infection and be suited to prevent relapse. A systematic approach is required to consider long-term follow-up within the context of adjunct treatments. Furthermore, we should not overlook topical formulation excipients aimed at promoting the penetration of the active ingredient (AI), namely alcohol, into the skin in multiarmed studies. It is also plausible that the eradication of commensal bacterial flora can greatly affect the resolution of symptoms, but equally the reoccurrence rate. The development of resistance to terbinafine [19] has led to in vitro susceptibility testing [20].

A consensus on the methodology of future trials would allow direct comparison across treatments for tinea pedis and provide clinicians with reproducible measures of the efficacy of different treatments, including newer experimental drugs, that would inform clinical practice. The efficacy of adjunct treatments demands further research. It is commonly advised by clinicians to maintain good foot hygiene, keep feet dry and wear textiles that wick moisture away from tinea-prone areas. None of the studies included in this systematic review investigated the role of these adjunct treatments and practices.

There is a large body of data demonstrating that currently approved pharmaceuticals increase the likelihood of successful treatment over a placebo, including this systematic review. Many now question the need for placebo-controlled trials when there is known benefit of many classes of anti-fungal agents suited to the treatment of tinea pedis. From a study design point of view, it enables conventional meta-analysis [18]. However, the network meta-analysis methodology enables comparison between studies without a consistent reference trial arm (placebo). The authors support stopping the use of the placebo arm from an ethical standpoint.

This review demonstrated the superiority of both topical allylamines and azoles over a placebo, and the superiority of terbinafine over itraconazole. The understanding of the relative effect of the foot environment, formulation and adjunct treatments is still poorly understood.

## Figures and Tables

**Figure 1 jof-08-00351-f001:**
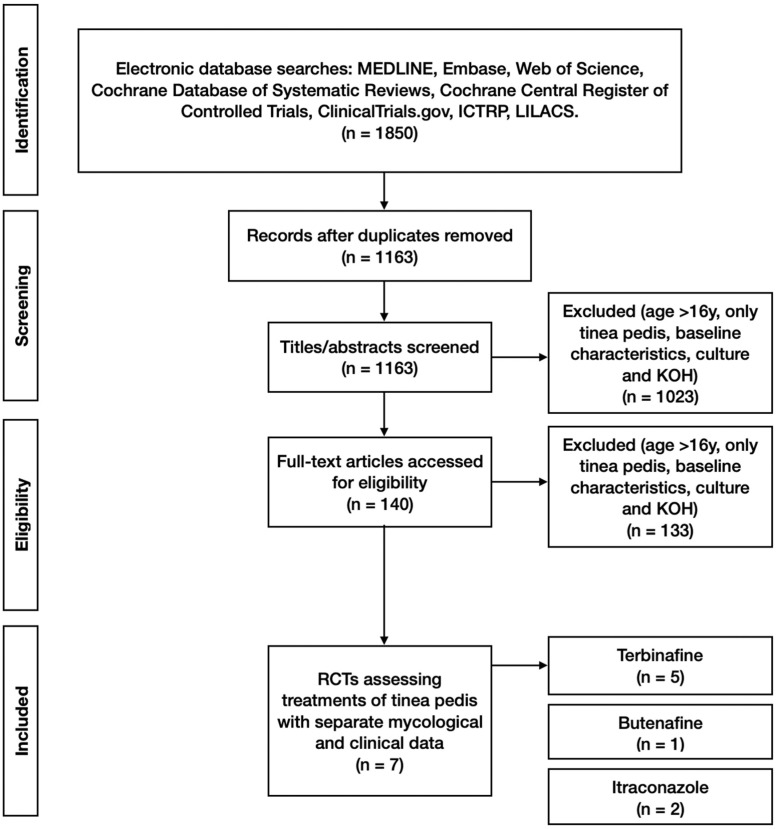
PRISMA (Preferred Reporting Items for Systematic Reviews and Meta-Analyses) flow diagram.

**Figure 2 jof-08-00351-f002:**
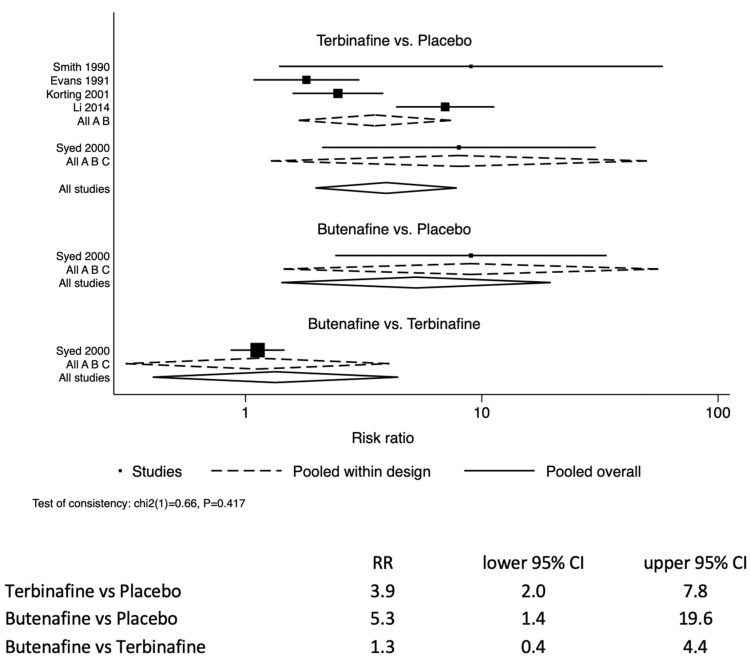
Forest plot of the likelihood of resolution of tinea pedis infections through topically applied treatments in randomised controlled trials, either placebo-controlled or multi comparator treatment arms.

**Figure 3 jof-08-00351-f003:**
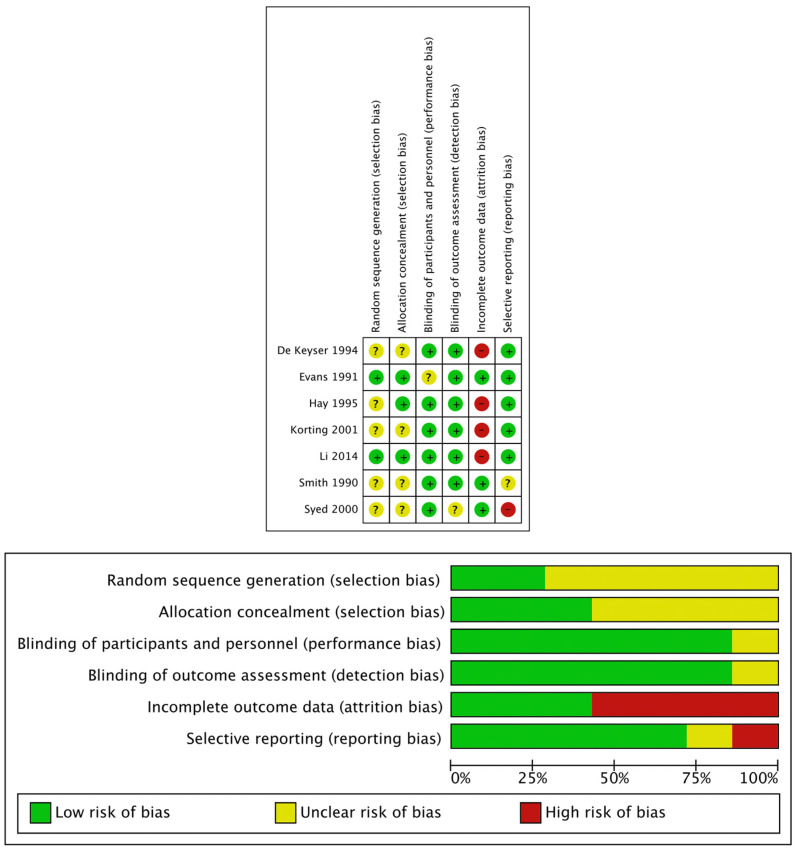
Risk-of-bias table for the tinea pedis randomised controlled trials.

**Table 1 jof-08-00351-t001:** Summary of the included randomised controlled trials.

	Smith et al., 1990		Evans et al., 1991		De Keyser et al., 1994		Hay et al., 1995		Syed et al., 2000			Korting et al., 2001		Li et al., 2014	
Arms	Placebo	Terbinafine	Placebo	Terbinafine	Terbinafine	Itraconazole	Terbinafine	Itraconazole	Placebo	Terbinafine	Butenafine	Placebo	Terbinafine	Placebo	Terbinafine
Number of Patients	10	10	20	28	184	182	88	100	20	20	20	35	35	145	145
Number of Patients with Mycology Data	10	10	20	27	51	66	65	64	20	20	20	35	35	122	115
Follow-Up (weeks)	6		6		8		16		4			8		6	
% Male	80		66.7		60.2		53.7		100			80		66.9	
Average age	29	40	47.7	46.5	40.6	38.5	46.6	45.7	35.2	34.8	35.9	46	42	34.1	35.3

## Data Availability

We confirm that the manuscript is an honest, accurate, and transparent account of the study being reported; that no important aspects of the study have been omitted; and that any discrepancies from the study as planned (and, if relevant, registered) have been explained.

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
