# Peer review of "Consensus for the Treatment of Tinea Pedis: A Systematic Review of Randomised Controlled Trials"

_jof, 2022, doi:10.3390/jof8040351_

Round 1

Reviewer 1 Report

Interesting article written in an accessible language. The approach to the topic is very scientific, but it is a pity that the literature review is limited to a rather small number of publications. In general, the article is a good overview of the issue along with data interpretation.

Main suggestions:

  • It seems to me justified to introduce a table summarizing all analyzed studies. In such a table, more detailed information about patients, therapy, cure, etc. can be given. This will make it easier to find out about the topic and will be a good complement to paragraph 4 in which there is an analysis of these studies.
  • I propose to completely abandon the network meta-analysis map. The number of studies and drugs used is too small for such an analysis to make sense. In my opinion, without any substantial loss to the article, figures 2 and 4 can be removed

Minor suggestions:

  • line 16: I suggest to use the full name here and the RCT abbreviation in brackets
  • line 35-37: names of dermatophyte species should be italicized
  • line 38-39: Currently, molecular methods are increasingly used, even those based on real-time PCR. One sentence can be added here about this, because the mere limitation to receiving culture is not enough
  • line 272: This is not a table, but rather a simple figure. In my opinion, this is not a figure that covers the subject completely. I propose to delete and keep only the text description, which is sufficient

Author Response

Thank you for your helpful comments on our manuscript. I will take each point in turn and explain how it has been addressed in this revision.

  • It seems to me justified to introduce a table summarising all analysed studies. In such a table, more detailed information about patients, therapy, cure, etc. can be given. This will make it easier to find out about the topic and will be a good complement to paragraph 4 in which there is an analysis of these studies.

Thank you for your comment. We have included Table 1, which summarises each study included in our review. This provides demographic details about the included patients as well as details of the treatment arms and the duration of these treatments. We are happy to put this table in further material at your discresion.

  • I propose to completely abandon the network meta-analysis map. The number of studies and drugs used is too small for such an analysis to make sense. In my opinion, without any substantial loss to the article, figures 2 and 4 can be removed

Thank you for your comment. We have removed these two figures and removed comments on meta-analysis.

Minor suggestions:

  • line 16: I suggest to use the full name here and the RCT abbreviation in brackets

Thank you for your comment. We use the full term ‘randomised controlled trials’ and then give the abbreviation (RCT) afterwards. After that, we refer to them as ‘RCTs’.

  • line 35-37: names of dermatophyte species should be italicised

Thank you for your comment. We have italicised these names.

  • line 38-39: Currently, molecular methods are increasingly used, even those based on real-time PCR. One sentence can be added here about this, because the mere limitation to receiving culture is not enough

Thank you for your comment. We have added in a sentence on this.

  • line 272: This is not a table, but rather a simple figure. In my opinion, this is not a figure that covers the subject completely. I propose to delete and keep only the text description, which is sufficient

Thank you for your comment. We have removed this table and kept the text description.

Reviewer 2 Report

The authors are advised to consider the following suggestions to improve the quality of the manuscript.

  1. The introduction section should be elaborated with a detailed background which helps to build the hypothesis and need of the present study/investigation.
  2. It is suggested to include the previous important findings of a similar line of research in the introduction section and highlight the current research gap in this area. How the present investigation is helpful to address this research gap or provide value addition in this research area should be added in the revised manuscript. Although, the author has addressed related issues in the introduction section. But it should be reorganized in a better way to make it more interesting for the reader. Generalized/obvious information needs to be deleted from the introduction/background section.
  3. In my opinion, the methodology section should be divided into different subsections for a quick understanding of the readers/viewers.
  4. The discussion section needs to be elaborated with suitable citations of the recent research. 
  5. The conclusion section needs to be incorporated with the inclusion of elaborative future directions in this area of research. 

Author Response

Thank you for your helpful comments on our manuscript. I will take each point in turn and explain how it has been addressed in this revision.

1. The introduction section should be elaborated with a detailed background which helps to build the hypothesis and need of the present study/investigation.

Thank you for this comment. We have added in detail to the end of the fourth paragraph in the introduction to highlight the conflict in the field surrounding the efficacy of allylamines versus azoles. We also make the point that we add benefit to the field by being critical of previous trials that have a high risk of bias, thereby providing an up-to-date view of the field and one based on only the best quality evidence.

2. It is suggested to include the previous important findings of a similar line of research in the introduction section and highlight the current research gap in this area. How the present investigation is helpful to address this research gap or provide value addition in this research area should be added in the revised manuscript. Although, the author has addressed related issues in the introduction section. But it should be reorganised in a better way to make it more interesting for the reader. Generalised/obvious information needs to be deleted from the introduction/background section.

Thank you for this comment. We summarise the results of previous systematic reviews on this topic in the fourth paragraph of the introduction, highlighting the inconsistencies and the need for clarity. We have split the fourth and fifth paragraphs and added detail to the end of the fourth paragraph, making this argument clearer for the reader. We have removed details around the clinical manifestations of tinea pedis and the mechanism of anti-fungal agents as this is expected knowledge for the readership.

3. In my opinion, the methodology section should be divided into different subsections for a quick understanding of the readers/viewers.

Thank you for this comment. We have added subsections to the methodology section to make this clearer for the reader.

4. The discussion section needs to be elaborated with suitable citations of the recent research. 

Thank you for this comment. We cite recent systematic reviews which cover this topic. We have added a citation around a comment that clinical manifestations of tinea pedis can persist after mycological eradication of the infection. We have also added a citation on PCR techniques that allow the quantification of mycological infection.

5. The conclusion section needs to be incorporated with the inclusion of elaborative future directions in this area of research. 

We have added a subtitle in the discussion section to demarcate the ‘future perspectives’ section clearly. We discuss standardisation of methodology in RCTs and examples of how this could be achieved, the importance of long-term follow-up to investigate recurrence of tinea pedis, the importance of studies into adjunct treatments such as footwear and hygiene, and comment on the controversy over the ethics of placebo-controlled trials in the treatment of tinea pedis.

Round 2

Reviewer 1 Report

Interesting article. Congratulations!

Reviewer 2 Report

The revised manuscript (jof-1635046) improved well after incorporation of given suggestions. In my opinion, present form of manuscript should be considered for publication.